# Research of an Unmanned Aerial Vehicle Autonomous Aerial Refueling Docking Method Based on Binocular Vision

**Kun Gong** [1] , **Bo Liu** [2], **Xin Xu** [1], **Yuelei Xu** [1,*], **Yakun He** [2], **Zhaoxiang Zhang** [1] **and Jarhinbek Rasol** [1]

[1] Unmanned System Research Institute, Northwestern Polytechnical University, Xi'an 710072, China; gongkun@mail.nwpu.edu.cn (K.G.); xu_xin99@mail.nwpu.edu.cn (X.X.); zhangzhaoxiang@nwpu.edu.cn (Z.Z.); jarhinbek.r@mail.nwpu.edu.cn (J.R.)

[2] Chinese Aeronautical Establishment, Beijing 100012, China; liob031@avic.com (B.L.); heyk003@avic.com (Y.H.)

* Correspondence: xuyuelei@nwpu.edu.cn

**Abstract:** In this paper, a visual navigation method based on binocular vision and a deep learning approach is proposed to solve the navigation problem of the unmanned aerial vehicle autonomous aerial refueling docking process. First, to meet the requirements of high accuracy and high frame rate in aerial refueling tasks, this paper proposes a single-stage lightweight drogue detection model, which greatly increases the inference speed of binocular images by introducing image alignment and depth-separable convolution and improves the feature extraction capability and scale adaptation performance of the model by using an efficient attention mechanism (ECA) and adaptive spatial feature fusion method (ASFF). Second, this paper proposes a novel method for estimating the pose of the drogue by spatial geometric modeling using optical markers, and further improves the accuracy and robustness of the algorithm by using visual reprojection. Moreover, this paper constructs a visual navigation vision simulation and semi-physical simulation experiments for the autonomous aerial refueling task, and the experimental results show the following: (1) the proposed drogue detection model has high accuracy and real-time performance, with a mean average precision (mAP) of 98.23% and a detection speed of 41.11 FPS in the embedded module; (2) the position estimation error of the proposed visual navigation algorithm is less than $\pm0.1$ m, and the attitude estimation error of the pitch and yaw angle is less than $\pm0.5°$; and (3) through comparison experiments with the existing advanced methods, the positioning accuracy of this method is improved by 1.18% compared with the current advanced methods.

**Keywords:** object detection; pose estimation; binocular vision; autonomous aerial refueling; UAV

## 1. Introduction

Unmanned aerial vehicles (UAVs) have recently been widely used in a variety of fields thanks to advances in modern UAV technology. However, UAVs are severely limited in their potential applications because of their insufficient fuel capacity and short endurance, which prevents them from completing long-duration, long-distance missions. The distance, retention time, activity space, and payload of UAVs may, however, be enormously increased thanks to a technology called unmanned aerial vehicle autonomous aerial refueling (UAV-AAR), without essentially altering the original architecture of the aircraft [1–3].

There are two main types of refueling for aircraft: hard-tube (U.S. Air Force) and hose-tube (U.S. Navy, Chinese Air Navy, and other nations) [4]. For hose-tube refueling, there are five main steps: rendezvous, formation, docking, delivery, and exit, with docking being the most important and the determining factor in whether the entire procedure is successful or unsuccessful [5]. Therefore, obtaining an accurate relative pose between the receiver aircraft and the drogue becomes crucial for real-time navigation during the UAV-AAR docking process [6].

The sensor error in the docking process of UAV-AAR is at least below the centimeter level, but the current conventional satellite navigation, as well as differential satellite

navigation with higher accuracy, cannot meet this requirement. Therefore, visual navigation has emerged as the predominant approach to address this challenge and has garnered significant attention from researchers worldwide. The visual navigation approach for the UAV-AAR docking mission is depicted in Figure 1, and it primarily uses a coordinate solution to resolve the relative coordinate relationship between the oil receiving rod and the drogue.

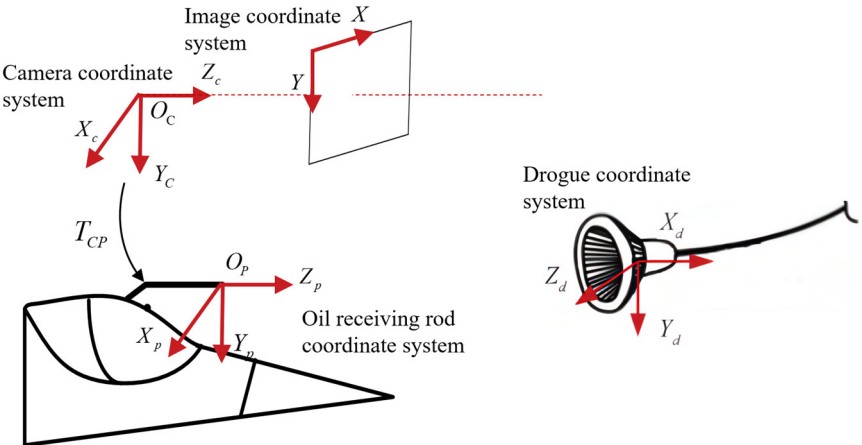

**Figure 1.** Schematic diagram of visual navigation coordinate system conversion.

VSLAM (Visual Simultaneous Localization and Mapping) methods, such as ORB-SLAM [7] and VINS [8], have been widely applied in UAV visual navigation. These methods utilize the perception of the surrounding environment to achieve real-time UAV localization [9]. However, in the context of UAV-AAR, VSLAM methods are not suitable. The main reason is that UAV aerial refueling visual navigation requires determining the relative pose information between the receiver aircraft and the drogue located on the tanker aircraft as guidance information. VSLAM methods, on the other hand, can only output the position of the UAV itself and cannot provide the relative pose information of the receiver aircraft with respect to the drogue. Currently, the primary core technologies used in visual navigation for UAV aerial refueling are drogue recognition, tracking, and pose estimation.

False detection, missed detection, and lost following are common problems because of the drogue's considerable swing and jitter in the air and the UAVs' quick flying speed. For instance, many researchers have introduced the YOLO family of algorithms [10–12] and EfficientDet [13] proposed by Google, among others, to the detection of drogues. Although the method is more robust, the real-time performance is subpar and cannot meet the high frame rate requirements in AAR tasks. Passive vision and active vision navigation are currently the two basic methods used to estimate the drogue pose in UAV-AAR visual navigation. Passive vision utilizes the inherent structural characteristics of the drogue to model and estimate the pose by extracting its circular features. Although this method can directly obtain the drogue pose, the calculation process is difficult and susceptible to environmental interference; in contrast, the active vision navigation method can more easily extract the drogue features through the use of optical auxiliary markers and is not susceptible to environmental interference.

To address the above problems, this paper proposes a UAV-AAR visual navigation system based on deep learning and binocular vision in three aspects: lightweight detection model improvement and feature extraction capability enhancement, multiscale adaptive object tracking, and high-precision drogue relative pose optimization. The main contributions of this article are as follows:

(1) In this paper, we propose a lightweight drogue feature extraction and matching method based on deep learning. Our approach combines object detection networks and artificial operators, utilizing the characteristics of the drogue and optical-assisted marker features for enhanced performance. This improves the scale adaptation and

real-time performance of the object detection network and successfully realizes the feature extraction and matching of the drogue.

(2) We propose a binocular vision-based drogue pose estimation method for the UAV-AAR vision navigation problem, which can effectively complete drogue pose estimation and provide accurate navigation information. In addition, a visual reprojection-based pose optimization method is proposed, which can further improve the accuracy and robustness of the drogue pose estimation algorithm by using more feature points.

(3) By constructing the UAV-AAR vision simulation system and the semi-physical UAV-AAR simulation experimental environment, this paper completes the experimental verification of the whole UAV-AAR visual navigation method, such as drogue object detection, object tracking, and the pose estimation algorithm, and the experimental results verify the reliability and robustness of this paper's method.

The following is an organization of the study's key components. In Section 2 of this study, a succinct summary of pertinent studies on visual navigation for UAV-AAR is provided. In Section 3, we delve into the details of the visual navigation method proposed in this paper, providing a comprehensive theoretical framework for drogue object detection, object tracking, and pose estimation. In Section 4, the effectiveness and applicability of the proposed method are demonstrated through virtual simulation and semi-physical simulation experiments. Furthermore, a comparison is made between this method and other approaches to highlight its superiority. The conclusions of this research are outlined in Section 5, along with some suggestions for future work.

## 2. Related Works

The AAR technology of UAVs has evolved into an advanced technology that is highly sought after by many nations, attracting numerous scholars from around the world to study this issue. This section reviews pertinent studies on drogue detection and tracking, visual navigation for UAV-AAR, and the benefits and drawbacks of different methods concisely.

### 2.1. Drogue Detection and Tracking

The work related to AAR navigation of UAVs is mainly focused on vision because drogue identification and tracking is the most critical problem in AAR and has become the focus of researchers. Zhang et al. [14,15] used the original 2D image-based tracking method, introduced 3D information and fed it back to the tracking algorithm, eliminated candidate regions, reduced the error rate, and improved the accuracy of the tracking algorithm. Choi A J et al. [16] used a K-means clustering method to cluster the bounding boxes of objects in the training data to determine the size of the anchoring box and implemented drogue detection and tracking using a YOLOV3-based drogue detection model. Ma et al. [17] proposed a real-time embedded drogue detection method based on quantized convolutional neural networks. An optimized detection network based on a multi-receptive field backbone network and depth-separable residual prediction blocks was designed to achieve real-time performance detection of drogues. Duan et al. [18] used a hawk-eye-based biological color detection method to extract drogue regions and markers and selected a visual navigation method with different distances based on whether the relative position difference between all detected markers and the receiver on the $x$ axis was greater than a threshold value to estimate the pose of the drogue. Xu et al. [19] proposed a convolutional neural network (CNN) classifier using adaptive boosting (Adaboost) with an improved focal loss (IFL) function to detect drogues in complex environments and solved the sample imbalance problem in the training phase of the CNN classifier using the IFL function. Gao et al. [20] proposed a real-time drogue tracking algorithm which can complete various tasks such as prediction, detection, and tracking. A fusion strategy of integrated prediction, detection, and tracking is designed by introducing the Kalman filter (KF).

### 2.2. UAV Air Refueling Visual Navigation

In the field of passive vision-based aerial refueling visual navigation, C Martínez et al. [21] proposed a monocular vision-based AAR navigation method for UAVs using an image-alignment algorithm and Zhong et al. [22] used an object detection network to propose a monocular vision-based drogue relative position estimation method. Campa et al. [23] and Fravolini et al. [24] used machine vision to fuse the position information obtained by feature matching with GPS to achieve the relative position of the fueling and receiving aircraft. Ma [25] and Zhao et al. [26] extracted measurement features using the arc-level structural feature extraction algorithm of drogues and proposed a monocular vision-based relative pose measurement method for fueling and receiving UAVs. In the field of aerial refueling visual navigation based on active vision, researchers assisted visual navigation by artificially adding optical auxiliary markers, such as LED lights, color markers, band markers, etc., as shown in Figure 2a–c.

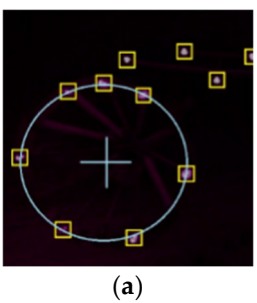 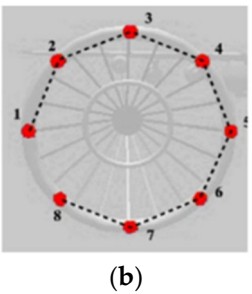 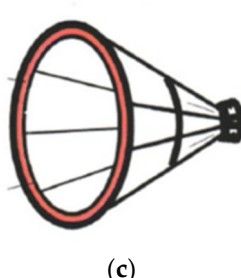

(**a**)  (**b**)  (**c**)

**Figure 2.** Marking types of optical-marker-assisted vision scheme for AAR with hose. (**a**) LED lights [27], (**b**) color markers [28], (**c**) band marker [29].

Luo et al. [30] proposed a binocular vision navigation method based on LED features, using a modified Harr wavelet transform to describe the feature points and estimate the pose of the drogue by matching the description vector of the feature points. Qin et al. [31] proposed a drogue detection algorithm based on binocular vision by adding artificial markers. Xie et al. [32] proposed a binocular vision-based proximity navigation method, which uses two cameras on the receiver to capture the images of optical markers on the end face of the refueling drogue to complete the matching of image points according to the pair of polar geometric constraints and uses the spatial circular fitting method to calculate the position and pose of the refueling drogue. To determine the position and pose of the refueling drogue, Pollini et al. [33] proposed a visual navigation method that uses infrared LEDs as optical marker points to determine the relative displacement and pose of the recipient by feature matching, constructed a refueling simulation testbed, and tested the effectiveness of the algorithm. Sun et al. [34] proposed using the RPNP method to estimate the drogue's pose and the comprehensive learning pigeon-inspired optimization (CLPIO) to optimize the rotation axis selection of RPnP, which keeps the algorithm from falling into the local optimum.

Generally, active vision image processing is simple, offers high real-time performance, and provides high accuracy at close range. On the other hand, the passive vision method, while having a simple structure and not requiring any additional auxiliary equipment, directly extracts features from the drogue itself, which is weaker than active vision in coping with changes in illumination and viewing angle, as well as local occlusion. Additionally, monocular vision utilizes the structural characteristics and known information of the drogue to recover the monocular scale and measure the relative pose of the drogue. Despite not being limited by the baseline distance and having a longer measurement distance compared to binocular vision, its accuracy and field of view are inferior to binocular vision methods for close-range navigation.

## 3. Methodology

To solve the problem of UAV-AAR, this paper fully combines the advantages of active vision and binocular vision to propose a visual navigation method and the method is mainly divided into four parts: drogue detection, feature extraction, object tracking, and pose estimation. In this paper, we use the single-stage lightweight drogue detection model to identify the drogue, obtain the position of the drogue in the image, and then continuously track the drogue by the N-fold Bernoulli probability-based adaptive fast-tracking algorithm. Finally, after feature extraction and triangulation of the spatial coordinates of the optical marker, we use the binocular vision-based pose estimation method to obtain the relative pose of the drogue and the camera. The overview of the method is shown in Figure 3.

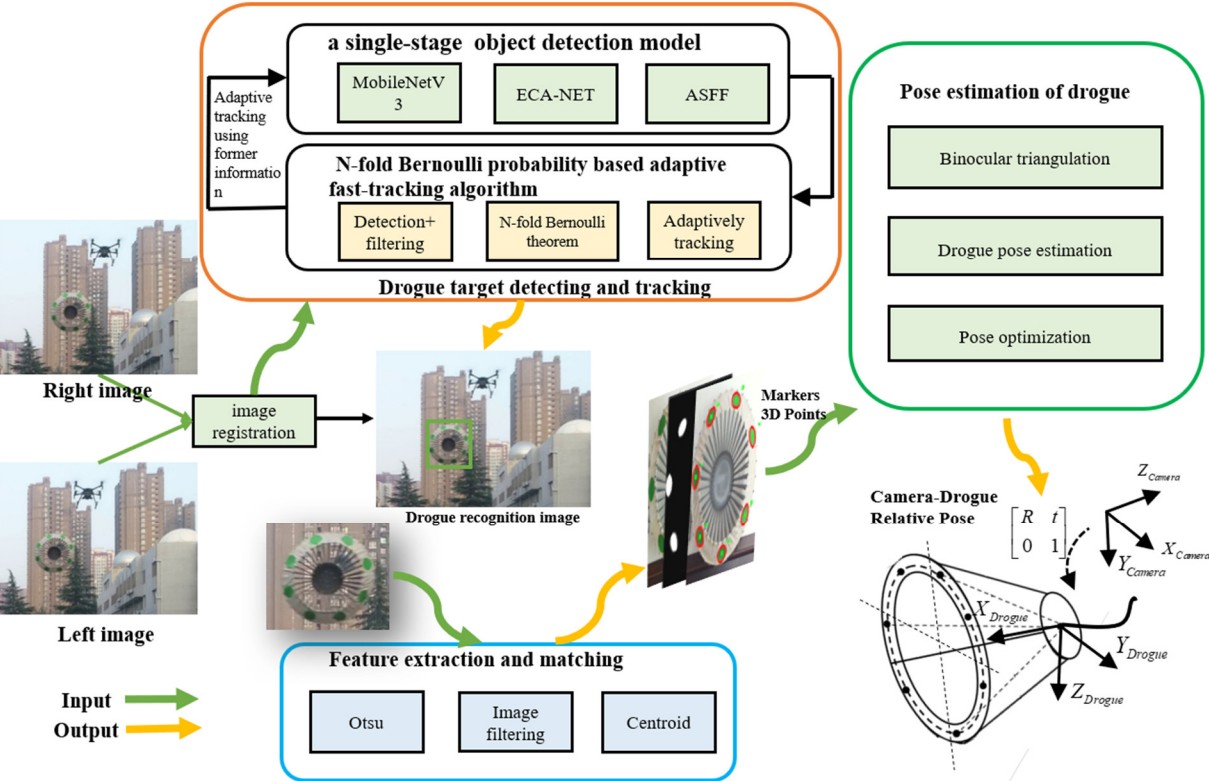

**Figure 3.** Framework of AAR visual navigation algorithm for UAVs.

### 3.1. Drogue Object Detection and Feature Extraction

During aerial refueling, the scale range of drogues is widely distributed, and the sizes range from small targets at the far end to large targets at the near end. Therefore, the object detection algorithm must be able to detect large and small targets at the same time, with strong adaptability to scale; in addition, in the aerial autonomous refueling mission, false detection may cause more serious consequences. As a result of the high speed and image frame rate of UAVs, it is imperative that the object detection algorithm used for UAV-AAR has high real-time performance and a low false detection rate. Therefore, the drogue object detection and feature-extraction algorithms need to be highly accurate and efficient to meet these requirements.

To address the aforementioned issues and achieve high accuracy while ensuring fast detection speed, we propose a deep learning-based feature extraction and matching method for drogues. Our approach is based on the advanced achievements of the YOLO object detection framework [10–12] in the field of object detection and can be divided into two main steps:

(1)    Accurate identification of drogue targets using an object detection network.

(2)    Extraction of the features of optical markers using traditional manual operators and matching to infer the 3D coordinate information of the optical marker.

In this paper, we propose a method to address the challenge of detecting binocular images simultaneously. We utilize an image-alignment method to encode and decode the drogue recognition results of binocular images simultaneously. To achieve quick and accurate detection of multiscale drogue targets in aerial refueling scenarios, this paper proposes the use of MobileNetV3 [35] as the backbone network. MobileNetV3 stands as the most recent milestone in the MobileNet series, building upon the depth-separable convolution of V1 and the residual structure of the linear bottleneck found in V2. Through the innovative design of network architecture search and lightweight feature extraction modules, it achieves a balance of high accuracy and low computational complexity. Additionally, we introduce a lightweight channel attention mechanism, ECA-NET [36], to replace the SE-NET [37] in the original MobileNetV3 network. This is because the ECA attention mechanism not only solves the channel compression and dimensionality reduction problem of the SE attention mechanism but also efficiently implements local cross-channel interaction using one-dimensional convolution. This enables it to obtain the importance of each feature channel and suppress some unnecessary feature information, which ultimately leads to superior speed and accuracy compared to the SE attention mechanism.

In addition, to improve the scale adaptability of the network, we introduce an adaptive spatial feature fusion method (ASFF) [38], which can adaptively change the spatial weights of feature fusion at different scales and improve the scale invariance of features by learning to filter conflicting information in the spatial domain to suppress inconsistent features, thus improving the feature fusion capability of the network for different scale targets. The structure diagram of its lightweight drogue object detection network is shown in Figure 4.

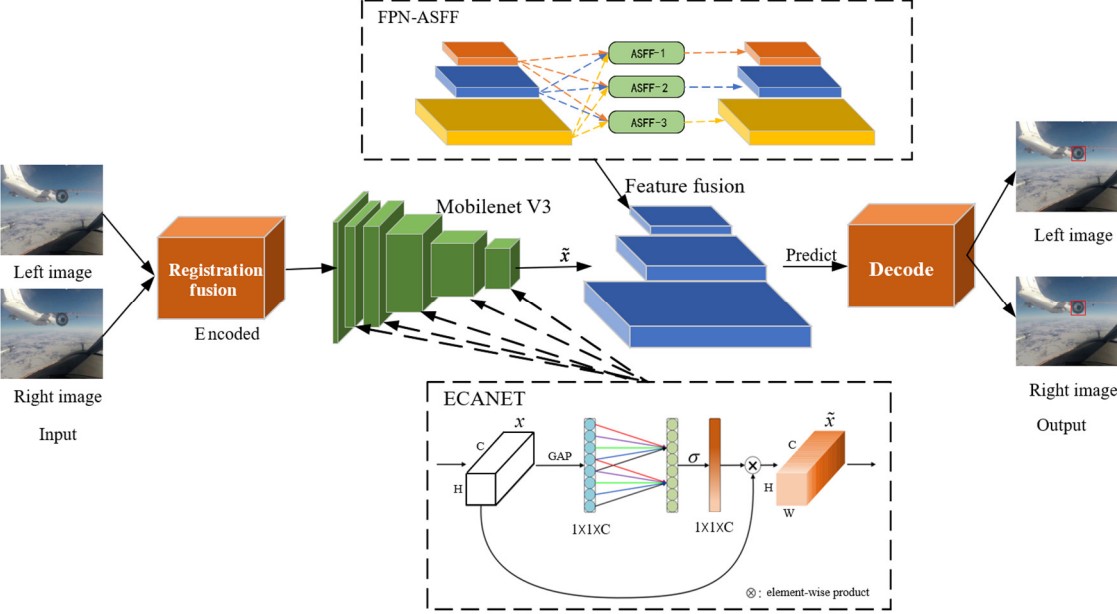

**Figure 4.** Structure of single-stage lightweight drogue object detection network.

After obtaining the location information of the drogue target in the image through the above object detection network, i.e., the drogue target region, RGB three-channel separation of the ROI (Region of Interest) is first performed, and the three-channel subtraction is performed to obtain the single-channel image containing the marker. Then, the rough binary image is obtained by the threshold segmentation through the Otsu method, after which the noise is eliminated by the median filtering and morphological operations to obtain the segmented image. Feature extraction of the optical marker is accomplished by the center-of-mass method, and then the feature matching of the left and right camera

images is performed to finally obtain the 3D coordinates of the optical marker in the camera coordinate system. The flow chart of the optical marker-based feature extraction and matching algorithm is shown in Figure 5.

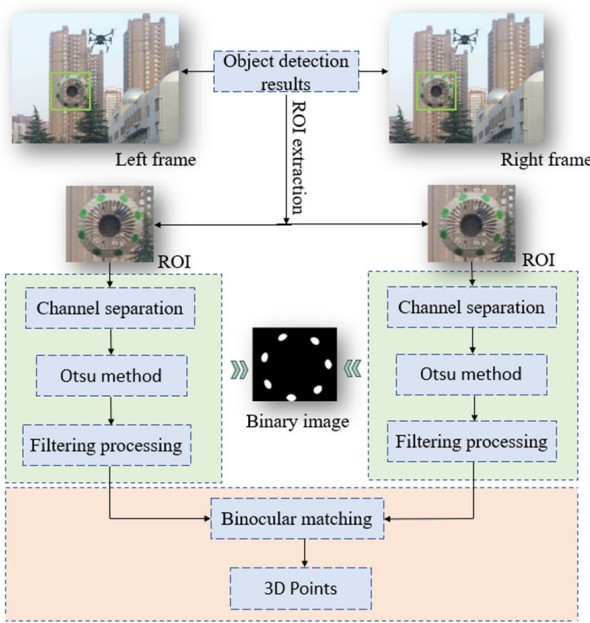

**Figure 5.** Feature extraction and matching algorithm based on optically assisted markers.

### 3.2. Drogue Object-Tracking Algorithm

Given the characteristics of a single tracking object, such as a wide range of scales and a significant amplitude of drogue oscillation in UAV aerial refueling missions, we have introduced an adaptive fast-tracking algorithm based on *n-weight Bernoulli*. This algorithm exhibits higher accuracy and stronger robustness when compared to other object-tracking algorithms. Additionally, it has a high inference speed, ensuring its use in embedded devices, as detailed in Rasol J's previous work [39].

### 3.3. Optical-Marker-Assisted Visual Navigation Method

#### 3.3.1. Binocular Triangulation Measurement Method

Binocular stereo vision is based on the principle of parallax. According to the principle of triangle similarity, the image coordinate system is transformed into the world coordinate system. This paper mainly discusses the method of parallel binoculars to obtain three-dimensional coordinates. The simple stereo parallel binocular imaging principle is shown in Figure 6.

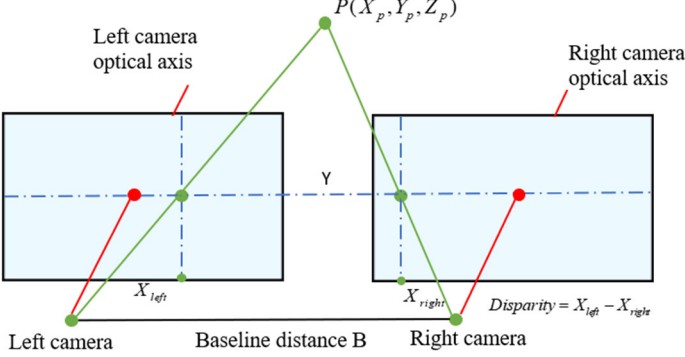

**Figure 6.** Stereo parallel binocular imaging schematic.

The baseline distance $B$ refers to the actual distance between the optical centers of the two cameras. The camera optical axis is a straight line through the center of all the lenses of the lens group. The disparity $D$ is the distance between the two points on the left and right graphic planes. Now, assume a point in space $P(X_p, Y_p, Z_p)$. It is in the image coordinates of the two cameras for points $P_{Left}(X_{Left}, Y)$ and $P_{Right}(X_{Right}, Y)$. Assume that intrinsic and extrinsic parameters of the two cameras are exactly the same, and the focal length of the two cameras is $f$, with the left optical center OL as the origin of the coordinates. According to the triangular similarity principle [40] of the image and object planes:

$$\begin{cases} X_{Left} = f\frac{X_p}{Z_p} \\ X_{Right} = f\frac{(X_p - B)}{Z_p} \\ Y = f\frac{Y_p}{Z_p} \end{cases} \tag{1}$$

The disparity $D = X_{Left} - X_{Right}$, so the three-dimensional coordinates of point $P$ in the camera coordinate system can be calculated.

$$\begin{cases} X_p = \frac{B \cdot X_{left}}{D} \\ Y_p = \frac{B \cdot Y_{left}}{D} \\ Z_p = \frac{B \cdot f}{D} \end{cases} \tag{2}$$

Therefore, $f$ is the camera focal length, and we only need to obtain the coordinates of the left camera image plane point in the binocular image and the coordinates of the point on the corresponding right camera image plane to obtain the disparity $D$. Then, we can determine the three-dimensional coordinates of the point $P$ in the camera coordinate system. Therefore, by adding optical markers on the refueling drogue, we can calculate the disparity $D$ by extracting the image coordinates of the left and right target points and performing feature matching to obtain the three-dimensional coordinates of each marker point.

3.3.2. Drogue Pose Estimation Method Based on Binocular Vision

Because the center of the drogue is used for refueling, it is not possible to set optical auxiliary markers at the center point of the drogue directly, so the 3D coordinates of the center of the drogue cannot be obtained directly by binocular vision measurement. Therefore, based on this principle, this paper proposes a method for measuring the coordinates of the center point of the drogue using an optical auxiliary marker. We assume that the three peripheral marker points $P1(X1, Y1, Z1)$, $P2(X2, Y2, Z3)$, and $P3(X3, Y3, Z3)$ can be measured using binocular vision, the three-dimensional coordinates of the center of the drogue are $O(X, Y, Z)$, and the three optical markers are on the same circle with a known distance $R$ from the center point to the optical marker. The value of $R$ can be calibrated using known quantities. The structure of the measurement method is schematically shown in Figure 7.

With $P1$, $P2$, and $P3$ as the centers of the spheres and $R$ as the radius, the three spherical equations can be obtained as shown in Equation (3).

$$C_i : (x - x_i)^2 + (y - y_i)^2 + (z - z_i)^2 = R^2 (i = 1, 2, 3) \tag{3}$$

Any two spheres intersect in the plane $M_{12}$, $M_{13}$, $M_{23}$, and any two equations of (3) can be subtracted to obtain the plane equation $M_{12}$, $M_{13}$, $M_{23}$ as:

$$\begin{aligned} M_{ij} : 2(x_j - x_i)x + 2\left(y_j - y_i\right)y + 2(z_j + z_i)z \\ = x_j^2 - x_i^2 + y_j^2 - y_i^2 + z_j^2 - z_i^2 \end{aligned} \tag{4}$$

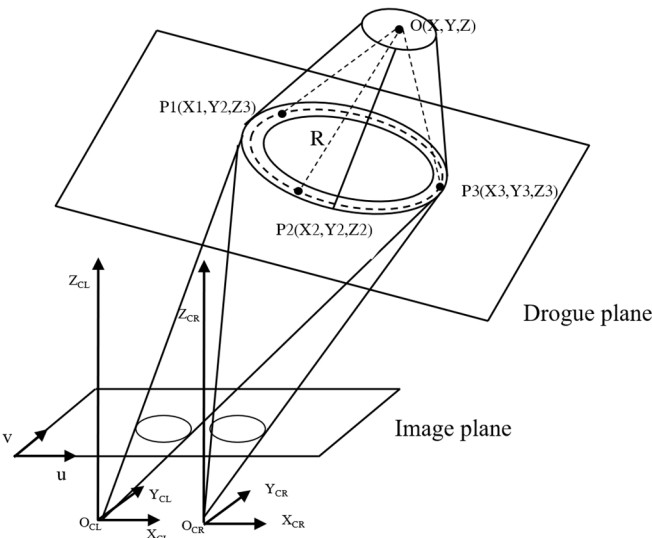

**Figure 7.** Schematic diagram of the structure of binocular vision-based drogue pose estimation method.

Therefore, the intersection of the three plane equations can be solved simultaneously to determine the three-dimensional coordinates of the center of the drogue as $O(X, Y, Z)$, and its matrix equation is obtained as follows:

$$\begin{bmatrix} 2(x_1 - x_2) & 2(y_1 - y_2) & 2(z_1 - z_2) \\ 2(x_1 - x_3) & 2(y_1 - y_3) & 2(z_1 - z_3) \\ 2(x_2 - x_3) & 2(y_2 - y_3) & 2(z_2 - z_3) \end{bmatrix} \begin{bmatrix} x \\ y \\ z \end{bmatrix} = \begin{bmatrix} x_1{}^2 - x_2{}^2 + y_1{}^2 - y_2{}^2 + z_1{}^2 - z_2{}^2 \\ x_1{}^2 - x_3{}^2 + y_1{}^2 - y_3{}^2 + z_1{}^2 - z_3{}^2 \\ x_2{}^2 - x_3{}^2 + y_2{}^2 - y_3{}^2 + z_2{}^2 - z_3{}^2 \end{bmatrix} \quad (5)$$

Because the three points $P1$, $P2$, and $P3$ must be coplanar, the three normal vectors of the plane $M_{12}$, $M_{13}$, $M_{23}$ are $\overrightarrow{P_1P_2}$, $\overrightarrow{P_1P_3}$, $\overrightarrow{P_2P_3}$, and the three points are in a plane, thus we obtain the following relationship.

$$\overrightarrow{P_1P_2} + \overrightarrow{P_2P_3} + \overrightarrow{P_3P_1} = \overrightarrow{0} \quad (6)$$

Therefore, the three normal vectors are linearly related, and their equations have infinite solutions, i.e., the rank of its augmentation matrix $R(A) < 3$. However, the three points are not colinear, so any two of the normal vectors must not be parallel, which can constitute a linear irrelevance group. Its augmentation matrix $R(A) > 1$, the rank of the augmentation matrix is 2, and by using the column principal Gaussian elimination method, we can find its general solution and set its system of equations. The general solution is:

$$u = ka + u_0 (k \in \forall) \quad (7)$$

Given that a is the general solution of the homogeneous system of equations, $u_0$ is the particular solution of the non-homogeneous system of equations, and $k$ is an arbitrary constant. By substituting the general solution into any spherical equation $C_i$, a quadratic equation about $k$ can be obtained. The straight line and the sphere generally intersect at two points, thus two solutions can be found. After obtaining the constant $k$, we substitute it into Equation (6). Because the drogue always faces the tanker, the solution with a larger distance z is selected as the positive solution. At this point, the three-dimensional coordinates of the center of the drogue in the camera coordinate system can be successfully found.

Suppose the camera coordinate system $\phi_a$ is rotated $\psi$ (pitch angle), $\theta$ (yaw angle), and $\phi$ around Y, X, and Z in turn to obtain the drogue coordinate system $\phi_b$, i.e., $\phi_a = R_{ab}\phi_b$, $R_{ab}$

is represented by Eulerian rotation as $R_{e_3,\phi}$, $R_{e_2,\theta}$, $R_{e_1,\psi}$, so the rotation transformation equation yields:

$$R_{ab} = \begin{bmatrix} \sin\phi\sin\theta\cos\psi - \cos\phi\sin\psi & \sin\psi\sin\theta\cos\phi - \cos\psi\sin\phi & \cos\theta\sin\psi \\ \cos\theta\sin\phi & \cos\theta\cos\phi & -\sin\theta \\ \sin\phi\sin\theta\cos\psi - \cos\phi\sin\psi & \sin\theta\cos\phi\cos\psi + \sin\phi\sin\psi & \cos\psi\cos\theta \end{bmatrix} \quad (8)$$

Additionally, by solving the super-definite linear equation, the equation of the space plane where the marked point is located can be found as:

$$ax + by + cz + d = 0, (c \neq 0) \quad (9)$$

where $a$, $b$, $c$, and $d$ are the coefficients of the plane equation set. Note that $k_0 = \frac{a}{m}$, $k_1 = \frac{b}{m}$, $k_2 = \frac{c}{m}$ (where $m = \sqrt{a^2 + b^2 + c^2}$). Then, from this, the unit normal vector of the space plane of the drogue where the marker point is located $\vec{n} = (k_0, k_1, k_2)$ can be obtained. If a point O of the plane is the origin, the normal vector $\vec{n}$ is the Z axis, and the mutual perpendicular vectors on the plane $\vec{i}$, $\vec{j}$ are the X and Y axes that establish the coordinate system of the drogue $\phi_b$. Therefore, we have $\phi_a = \left(\vec{i}, \vec{j}, \vec{n}\right)\phi_b$, so Equations (9) and (10) can be obtained from the pitch angle $\theta$ and yaw angle $\psi$ expressions as:

$$\theta = \text{acrsin}(-k_1) \quad (10)$$

$$\psi = \arcsin\left(\frac{k_0}{\cos\theta}\right) \quad (11)$$

Among them, the pitch angle is positive for upward deviation and negative for downward deviation; the yaw angle is positive for left deviation and negative for right deviation.

In the theoretical modeling analysis of the drogue mentioned above, the proposed method in this paper only needs three marker points to achieve drogue pose estimation, which is less than the four points required by the existing circle fitting methods [4], implying that the method in this paper is more resistant to interference in practice.

### 3.3.3. Visual Reprojection-Based Positional Optimization Method

To further discuss the question of whether the accuracy and robustness of the algorithm can be improved by increasing the number of markers, this paper proposes a visual reprojection-based drogue pose optimization method to further improve the accuracy and robustness of the drogue spatial coordinate measurement algorithm by using more markers, which is shown in Figure 8.

First, using the extracted spatial coordinates $O(X, Y, Z)$ of the center of the drogue, the image coordinates of the left and right camera of the center point of the drogue can be inversely solved. In addition, the intrinsic matrix of the left and right cameras can be obtained by camera calibration ($K_{left}$, $K_{right}$) so that the image coordinates of the center point of the drogue in the left and right camera can be inversely solved ($I_{left}$, $I_{right}$), namely:

$$\begin{cases} I_{left} = \begin{bmatrix} u_{left} \\ v_{left} \\ 1 \end{bmatrix} = K_{left} \begin{bmatrix} \frac{X}{Z} \\ \frac{Y}{Z} \\ \frac{Z}{Z} \\ 1 \end{bmatrix} \\ \\ I_{right} = \begin{bmatrix} u_{right} \\ v_{right} \\ 1 \end{bmatrix} = K_{right} \begin{bmatrix} \frac{X}{Z} \\ \frac{Y}{Z} \\ \frac{Z}{Z} \\ 1 \end{bmatrix} \end{cases} \quad (12)$$

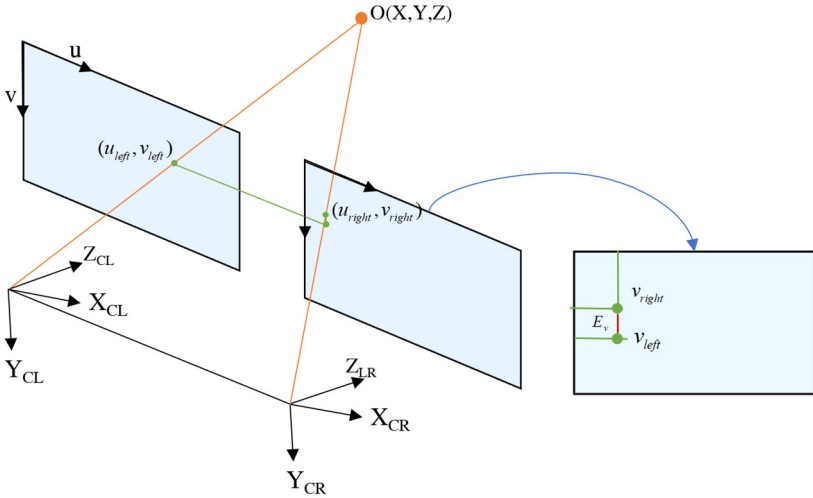

**Figure 8.** Schematic diagram of the visual reprojection method based on binocular vision.

Because the binocular images are stereo-rectified, the left and right camera images, after distortion elimination, correspond strictly in rows. The polar line constraint is utilized to align the opposite epipolar lines of the two images onto the same horizontal line. In other words, the vertical $Y$-axis coordinates of the images at the corresponding positions in the two images are identical. Therefore, the difference between the vertical coordinates of the image coordinates of the left and right objects, which are obtained by the inverse solving of the 3D coordinates of the drogue's center point, should be close to 0. The error term of its spatial coordinates is obtained as the difference between its longitudinal coordinates of the images after the inverse solution $E_v$:

$$E_v = \frac{1}{2}\left\|v_{left} - v_{right}\right\|^2 \tag{13}$$

Therefore, when the optical marking point is $\geq 3$, any three points can be taken to calculate the spatial coordinates of the center of the refueling drogue, and the error term $E_v$ can be calculated through the inverse coordinate solution. The smaller the error term $E_v$, the higher the precision of the spatial coordinates of the center of the refueling drogue will be. By this method, the three-dimensional coordinates of the center of the drogue with the smallest error is selected, solving the problem of when a point lowers the accuracy of the coordinates obtained due to the inaccurate extraction of features or noise interference, which is not available in other drogue pose estimation methods.

In addition, this paper proposes a nonlinear optimization method for distance $Z$. Using $\rho(E_v)$ as the loss function and distance $Z$ as the optimization variable, a nonlinear optimization equation is established, as shown in Equation (14).

$$L = \underset{Z}{\arg\min}\,\rho(E_v) \tag{14}$$

where $\rho$ is the Huber function [41] which reduces the effect of outliers. The derivative of the Huber function is shown in Equation (15), and the iterative optimization using the Gauss–Newton method [42] follows the specific iteration formula shown in Equation (16).

$$\frac{dL}{dz} = \begin{cases} E_v \frac{dEv}{dz} & (|E_v| \leq 1) \\[2mm] \frac{dEv}{dz} & (|E_v| > 1) \end{cases} \tag{15}$$

$$z^{k+1} = z^k - \left(\frac{dL}{dz}\right)^{-1} \cdot \mathrm{L}(z^k) \tag{16}$$

The measured value is used as the initial value $z^0$ for optimization. $z^{k+1}$ represents the estimated value after the $k + 1$ round of iterative optimization. In order to ensure the speed of the algorithm, the Newton–Gaussian method was used for five rounds of iterative optimization, to further obtain the optimal measured value of $Z$.

Using the above theoretical modeling of drogue pose estimation and the analysis of constructing a visual reprojection optimization method, the proposed binocular vision-based drogue pose estimation method can effectively complete the drogue pose measurement function and provide accurate navigation information and requires fewer marker points than the methods of Wang et al. [4]. and Xie et al. [29]., further improving the accuracy and robustness of the algorithm.

## 4. Experiment

### 4.1. Dataset

For the UAV aerial refueling scenario dataset construction problem, we developed a UAV aerial refueling data simulation experiment platform using a 1:1 custom drogue model and UAV platform, as shown in Figure 9, which mainly includes two sets of rotorcraft UAVs (refueler and recipient). The tanker UAV is DJI M210, and the receiver UAV is equipped with a Pixhawk flight control, AI module Nvidia Xavier AGX, and binocular camera. The binocular camera model is PXYZ-AR35-030T160, pixel size is $3.75 \times 3.75$ µm, resolution is $2560 \times 960$, focal length is 8 mm, baseline length is 12 cm. The AI module used is Nvidia Xavier AGX produced by NVIDIA Corporation, whose AI computing power reaches 32TOPS, the CPU model is 8-core NVIDIA Carmel Arm@v8.2, and the memory is 32 GB. The tanker UAV is equipped with a drogue with an outer radius of 33 cm. The surface of the drogue has 8 evenly distributed circular markings with a diameter of 10 cm.

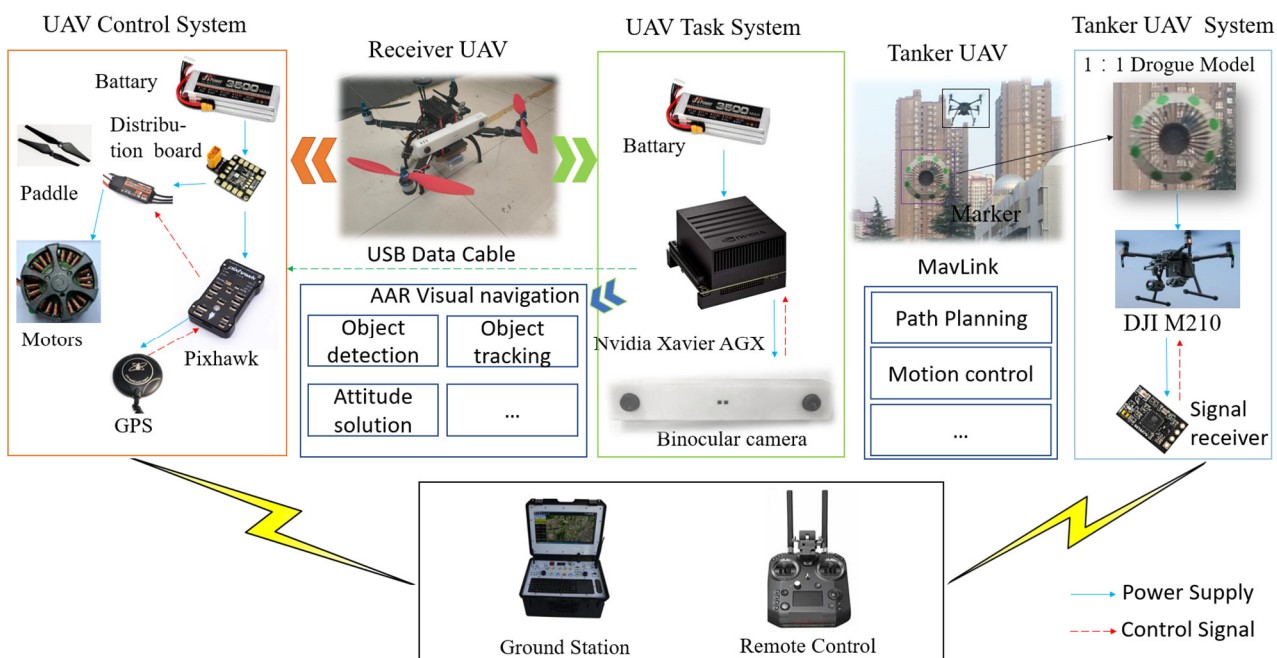

**Figure 9.** Air refueling data simulation experiment platform.

The drogue object dataset acquisition was carried out under many different backgrounds, different lighting conditions, and different distances, and 78 sets of videos were taken at a distance of approximately 2–40 m. In addition, we searched for drogue data of real refueling scenes on the internet, finding 33 sets of videos, and the dataset ratio of real scenes to semi-physical simulation scenes was approximately 4:1. By segmenting the video according to the isometric sampling method, the images containing drogues are selected, and the maximum possible number of drogue target images under multiple

scales and scenes are selected. Finally, a total of 14,298 images are selected as the object detection model dataset, and the dataset used in this paper covers the image size of the drogue target from approximately 2 m to 40 m, covering approximately 89 scenes. The location distribution diagram of the target size of the dataset is shown in Figure 10. It can be seen from the figure that the target location contained in the dataset basically covers all positions in the image, and the target size contained in the dataset is evenly distributed from small to large.

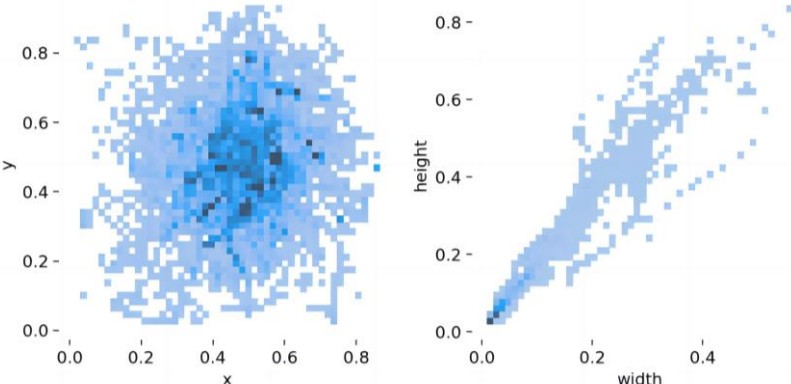

**Figure 10.** Size and location distribution of drogue targets in the dataset.

*4.2. Drogue Detection Experiment*

To verify the detection effect of aerial refueling drogues in various complex environments, this paper collects image data of real drogues in different environments. Data enhancement is performed using the mosaic data-enhancement method. The training set, validation set, and test set are trained with a ratio of 7:1:2 on a Windows 10 system equipped with an Nvidia GeForce RTX 3090 GPU. Comparison tests were performed with YOLOv5s [43], Efficientnet [13], Faster-RCNN [44], SSD [45], YOLOv8s [46], and YOLOv4 [12] models, and model accuracy and image processing time comparisons in the embedded Nvidia Xavier AGX are shown in Table 1, along with visualization as a tradeoff between mAP values and speed graphs, as shown in Figure 11.

**Table 1.** Performance comparison of different object detection models.

| Target Detection Model | Precision | Recall | mAP$_{0.5}$ | GFLOPs | Spend (s/img) |
|---|---|---|---|---|---|
| YOLOv8s [46] | 92.66% | 95.21% | 98.39% | 28.6 | 0.0387 |
| YOLOv5s [45] | 91.74% | 95.99% | 98.12% | 16.5 | 0.0311 |
| Effcientnet-B5 [13] | 91.07% | 93.48% | 96.68% | 9.9 | 0.0280 |
| Faster-RCNN [44] | 94.8% | 96.34% | 97.11% | 344.4 | 0.181 |
| SSD [45] | 91.79% | 92.03% | 95.33% | 31.0 | 0.0470 |
| YOLOv4 [46] | 92.05% | 92.05% | 95.59% | 52.0 | 0.0818 |
| ours-l | 93% | 99.5% | 98.23% | 9.3 | 0.0243 |
| ours-s | 94.29 | 91.49% | 96.29% | 6.0 | 0.0202 |

"Ours-l" refers to the backbone network using MobileNetV3-Large [35], while "ours-s" refers to MobileNetV3-Small [35]. As shown in Table 1 and Figure 11, in terms of speed, Faster RCNN has the slowest detection speed, while "Ours-s" has the fastest detection speed. In terms of detection accuracy, "SSD" has the lowest detection accuracy, with mAP0.5 only accounting for 95.33%. "YOLOv8s" has the highest detection accuracy, with mAP0.5 reaching 98.39%. Although mAP0.5 is not higher than that of the most advanced method at present (YOLOv8s), the real-time performance of our algorithm (Ours-l) is improved by 15.3 FPS without much loss in accuracy. Moreover, the aerial refueling mission demands higher real-time performance of the algorithm due to the UAV's high flight speed. By taking into account the scenario's applicability and the computing platform's processing

capability, our proposed method (Ours-l) exhibits superior real-time performance and level-pegging mAP0.5 compared to existing methods on the aerial refueling drogue dataset, boasting a 98.23% mAP0.5 and a detection speed of 41.11 FPS, with significant practical application potential.

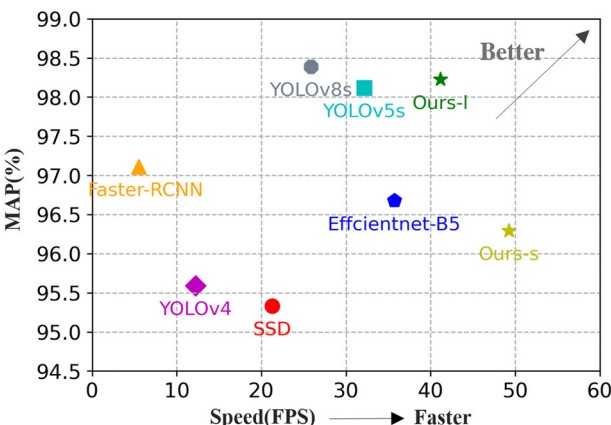

**Figure 11.** mAP and algorithm speed comparison chart of different methods.

In addition, to verify the actual effect and model generalization of the proposed object detection model in this paper, we conducted experiments in different backgrounds, different climatic conditions, and other complex situations. The experimental test results are shown in Figures 12–14.

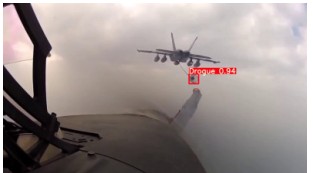 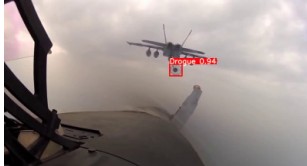 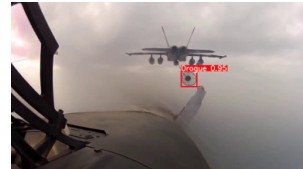 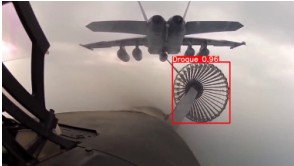

**Figure 12.** Drogue detection results under the cloud.

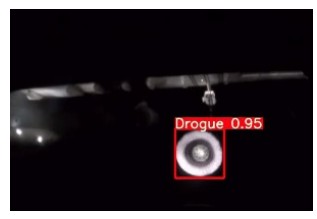 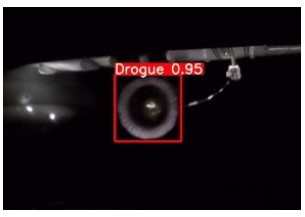 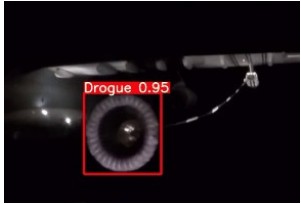 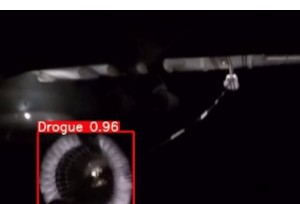

**Figure 13.** Drogue detection results under low-light conditions.

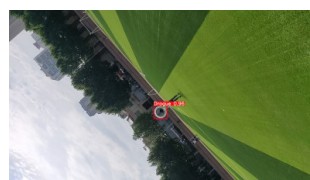 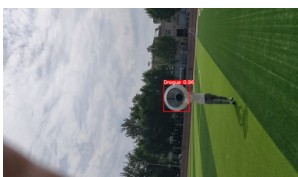 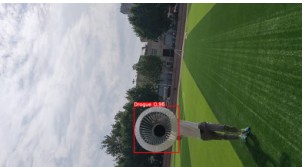 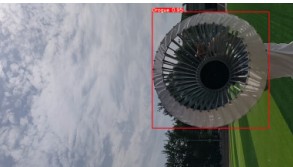

**Figure 14.** Ground drogue test results.

The effectiveness of the proposed lightweight drogue object detection network can be observed in Figures 12–14. The network is able to achieve successful detection with high confidence across various backgrounds and distances. These results demonstrate the strong generalization ability of the object model, making it suitable for practical applications.

### 4.3. UAV Docking Experiment

#### 4.3.1. View Simulation

To evaluate the effectiveness of the algorithm, we conducted simulation experiments in the Gazebo simulation environment within the ROS operating system. We used the topic communication method in ROS to facilitate the interaction between the algorithm and the camera sensor and employed time synchronization of multiple sensors to identify and track the drogue, ultimately determining its pose. The composition of the view simulation software system is shown in Figure 15.

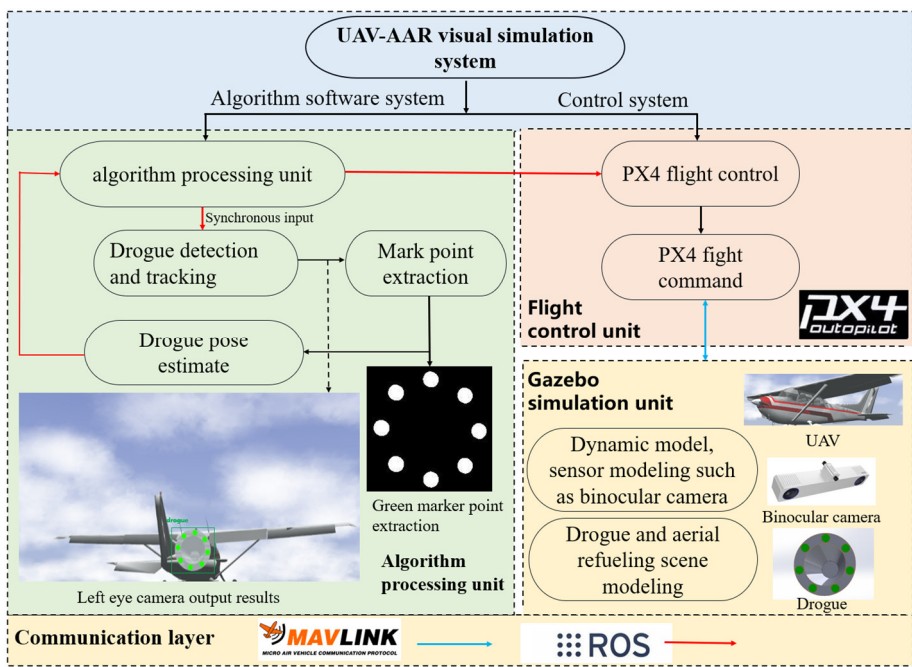

**Figure 15.** Schematic diagram of the composition of the aerial refueling visual navigation simulation software system for drones.

Through the sensor time-synchronization algorithm, the true value of the UAV drogue's pose during aerial refueling in the simulation system is output in real time and synchronized with the estimated value of the pose obtained by the visual navigation algorithm to obtain the drogue pose estimation error curve. In this paper, the error estimation curve of the trajectory of the UAV relative to the drogue when the UAV is docked is from 10 m to 1.5 m at a certain speed, as shown in Figure 16.

From Figure 16, it can be seen that as the distance increases, the drogue image feature area decreases, resulting in a more divergent error curve and a larger error fluctuation range, i.e., the farther the distance is, the larger the error; it can be seen that the position estimation accuracy of the proposed algorithm in the UAV aerial refueling visual navigation simulation system is less than $\pm 0.1$ m in the $X$, $Y$, and $Z$ axes, and the attitude estimation of its pitch angle and yaw angle is less than $\pm 0.5°$. Through the experimental analysis of the measurement accuracy of the above simulation system, it can be seen that the visual relative navigation system designed in this paper can meet the requirements of the UAV-AAR task. To further verify the effectiveness and advancement of the method in this paper, the visual navigation methods proposed by Ma et al. [5] and Wang et al. [4] were compared, and the receiver UAV still performed aerial refueling docking at a certain speed from a distance of 10 m, as shown in Figure 16.

In order to ensure the trajectory consistency of the three algorithms, we set the UAV aerial refueling docking path experiment in the virtual simulation system, and the ground truth and binocular images were released and recorded by using the topic function of ROS. In this way, the three algorithms could be tested under the same data, which not only

ensured the consistency of the environment but also ensured the synchronization of time. In addition, 8 optical auxiliary marks are set on the drogue of the tanker UAV with a radius of 10 cm. The running platform is ubuntu 18.04 and the CPU is i7-11700. The comparison of measurement results in the approaching process is shown in Figures 17 and 18.

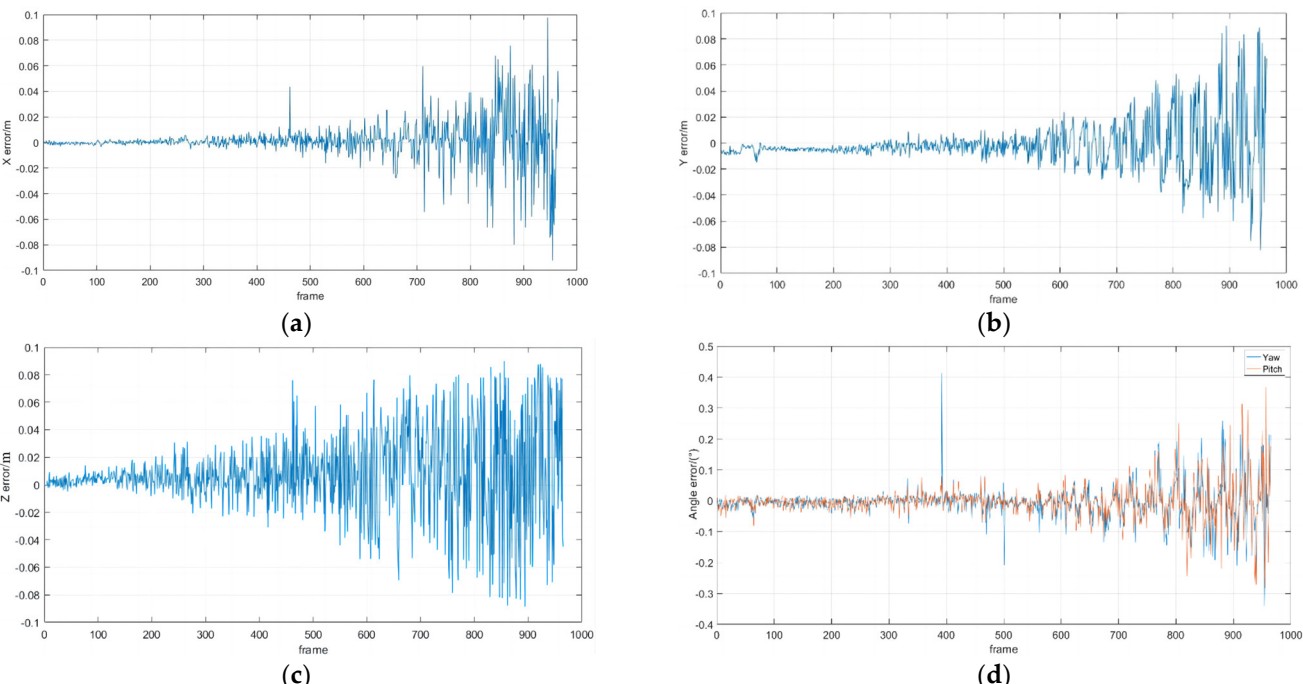

**Figure 16.** (**a**–**c**) show the error curves of the drogue coordinate system relative to the camera coordinate system *X*, *Y*, and *Z* axes. (**d**) show the error curves of yaw and pitch angle. The vertical axis represents the error, and the horizontal axis represents the sequence number of input images.

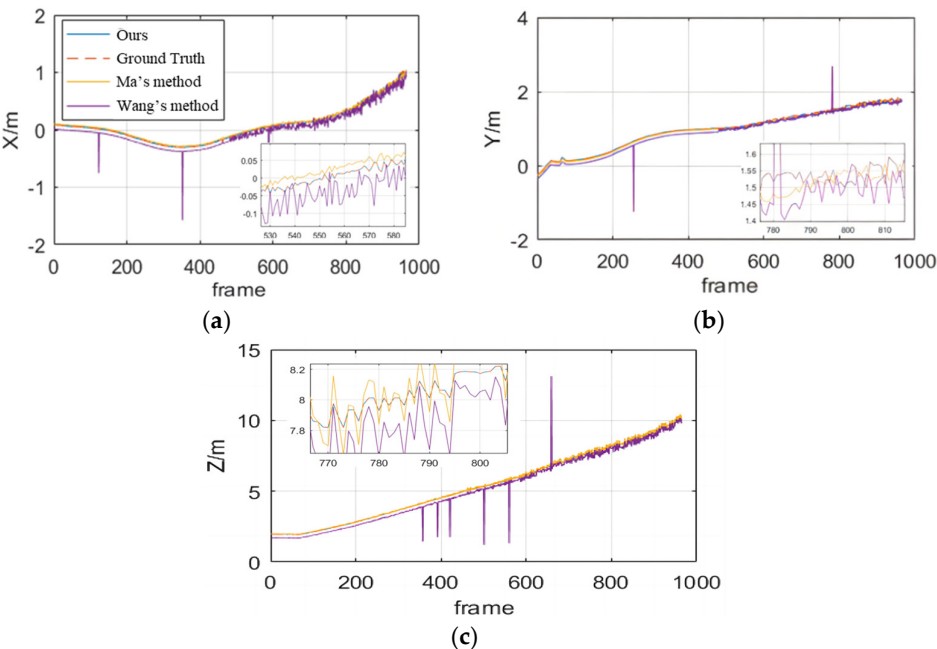

**Figure 17.** (**a**–**c**) represent the comparison curves between the estimated results and the ground truth of the *X*, *Y*, and *Z* axes' output by three algorithms during the UAV-ARR simulation process, respectively. The vertical axis represents the relative distance on the axis, and the horizontal axis represents the sequence number of input images.

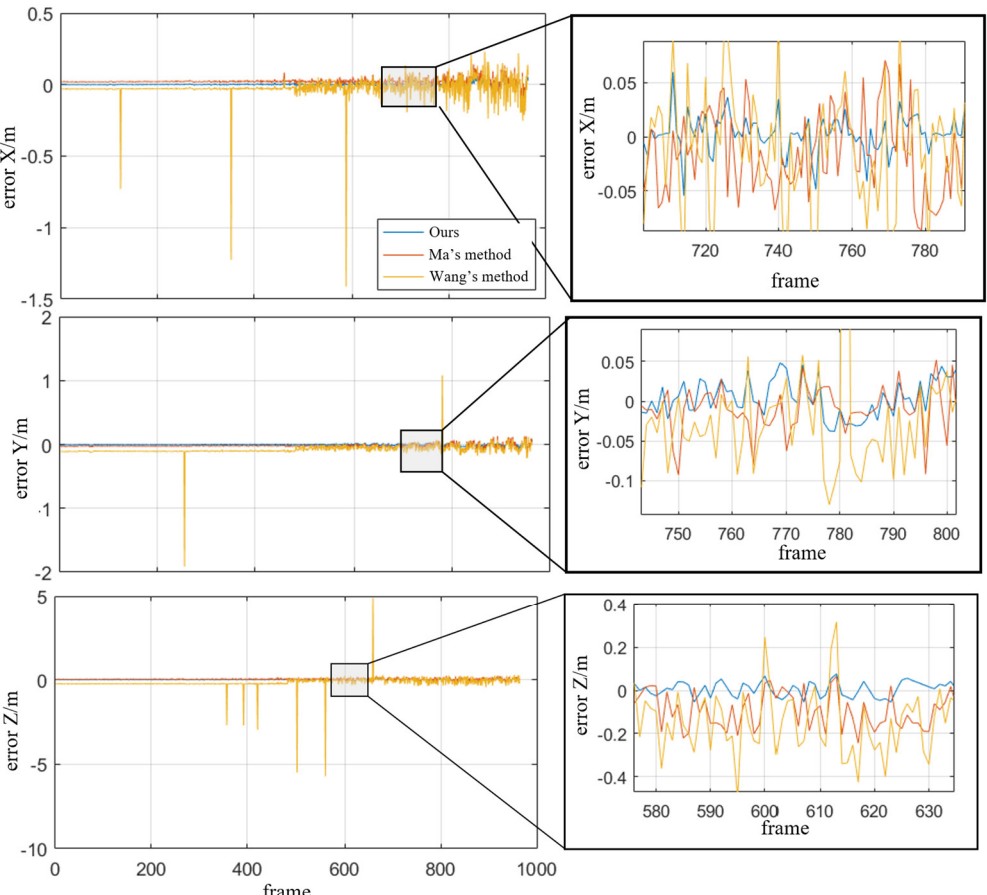

**Figure 18.** Error curves of the three algorithms on *X*, *Y*, and *Z* axes, respectively, during the UAV-ARR simulation process. The vertical axis represents error on the axis, and the horizontal axis represents the sequence number of input images.

As observed in Figure 17c, the *Z* axis increases as the image serial number increases, indicating that the distance between the receiver UAV and the drogue is becoming larger. Figure 18 further illustrates that the errors of the three algorithms also increase as the distance increases. From the comparison curves of the three algorithms, it can be seen that Wang's method has a large fluctuation range and is prone to large error values, indicating that it has poor robustness and the worst accuracy, with an average error of 0.2176 m. The accuracy of Ma's method is close to that of the method in this paper, and its average error is 0.0541 m. Compared with the existing typical algorithms, the proposed algorithm improves the error on the *XYZ* axis, and the average accuracy is 0.0214 m. The average accuracy is calculated as the average Euclidean distance between the measurement point and the truth point. Our algorithm avoids the occurrence of a wide range of chance errors by using the optimization method of visual reprojection and has a smoother trajectory. Our method is relatively more accurate and has the best performance in the simulation scene when comprehensively compared with existing algorithms. Based on the comprehensive comparison and analysis of existing algorithms presented above, it can be concluded that this paper's method demonstrates superior capabilities and effectiveness.

### 4.3.2. Semi-Physical Simulation Experiments

The experimental platform consists of a robotic arm simulation platform, a Jetson Xavier AGX embedded platform, a binocular camera, a laser rangefinder, and an isometric drogue model, with the laser rangefinder providing the true value. The focal length of the binocular camera is 8 mm, the resolution is 2560 × 960, and the baseline length is 12 cm. To verify the drogue position estimation method based on binocular vision as proposed in this

paper, the first step is to calibrate the binocular camera. To verify the spatial positioning of the drogue, we need to perform three steps of camera calibration, stereo correction, and stereo matching to achieve the spatial positioning of the drogue.

Due to the limitation of angle information acquisition in the actual environment, this paper only conducts experiments on the measurement error of the pose angle in the vision simulation and only conducts experiments on the positioning of the proximity drogue in the semi-physical simulation. To further demonstrate the effectiveness and accuracy of the method in this paper, it was compared with the visual navigation methods of Ma et al. and Wang et al. To ensure the accuracy of the comparison experiment and reduce the random error, a total of 100 sets of data were taken each time, and the average error of 10 sets was measured as the experimental results. In order to ensure the fairness of the experiment, we carried out the experiment under the same conditions. At the same time, we took 5 optical auxiliary markers with the same size and color and evenly distributed them in circles on the surface of the drogue. The hardware conditions such as camera parameters were consistent. The comparison of experimental results obtained is shown in Figure 18.

The experimental results depicted in Figure 19a,b demonstrate that the maximum errors of the *X* and *Y* axes in all three methods increase as the relative distance increases. Notably, the average error of the method proposed in this paper is comparable to, and in some cases smaller than, that of Ma et al. In contrast, Wang et al.'s method exhibits the largest average error and the lowest accuracy. All three methods show an increase in average error as the relative distance increases. However, when comparing the *Z*-axis error, it is evident that the error of the method proposed in this paper is significantly smaller than that of the other two methods. This result effectively demonstrates that our method exhibits higher algorithmic robustness and accuracy compared to existing methods.

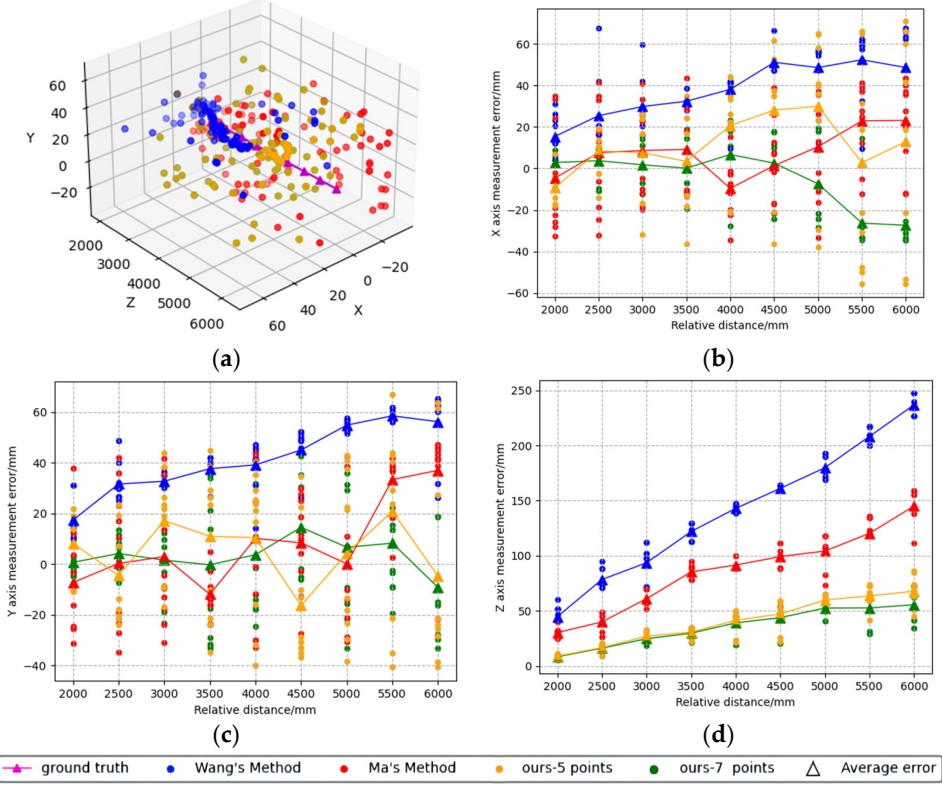

**Figure 19.** Position measurement error comparison chart in semi-physical simulation experiment. ((**a**) shows the comparison chart of different methods with the true value. (**b**–**d**) show the comparison chart of the error of different methods of the drogue coordinate system relative to the camera coordinate system *X*, *Y*, and *Z* axes. The vertical axis represents error on the axis, and the horizontal axis represents the relative distance between the camera and the drogue).

By examining the measurement results in Table 2, we can observe that the average error is noticeably smaller when there are 7 optical markers as compared to 5 markers. This is attributed to the visual reprojection-based optimization method, which reduces errors and avoids the occurrence of a wide range of errors. In addition, the average measurement error of the *X* axis is only 4 cm, the *Y* axis is 4 cm, and the *Z* axis is only 1.03%, which is better than the other two methods in terms of performance, and our algorithm can be run on the embedded platform to meet the real-time performance requirements.

**Table 2.** Table of experimental results of system positioning.

| Method | Location Error | | |
|---|---|---|---|
| | *X* Axis Error/mm | *Y* Axis Error/mm | *Z* Axis Error/Z |
| Wang's method | ≤70 | ≤65 | ≤3.95% |
| Ma's method | ≤50 | ≤50 | ≤2.39% |
| Ours-5 Points | ≤65 | ≤65 | ≤1.21% |
| Ours-7 Points | ≤40 | ≤40 | ≤1.03% |

## 5. Conclusions and Future Works

To ensure the accuracy of the model while considering the real-time performance requirements of practical applications, a deep learning-based drogue feature extraction and matching network are proposed, and a complete drogue dataset is constructed. Based on this, we further proposed a binocular vision-based pose estimation method and obtained accurate pose information of the drogue through spatial geometric relationships and visual reprojection. The results from the view simulation and semi-physical simulation show that the proposed algorithm provides relatively accurate relative pose information of the drogue in the aerial refueling vision navigation of UAVs. The proposed algorithm provides relatively accurate relative pose information of the drogue in UAV aerial refueling visual navigation and meets the real-time performance requirements, which has broad application prospects.

However, the proposed method still has some limitations. For example, due to the influence of the binocular baseline, the accuracy of pose estimation is higher at close distances (less than 10 m), while the error becomes larger at longer distances. Moreover, the pure vision method is prone to large errors once the target is lost or the drogue is blocked. In future research in this area, some end-to-end neural networks can also be introduced into the air refueling process to achieve the extraction of the outer circle of the drogue, and through the extraction of the outer circle, the measurement of the drogue pose under the condition of monocular vision can be achieved and can even achieve better results. Future aerial refueling visual navigation of UAVs should pay more attention to safety and stability considerations to prevent accidents when two aircraft cooperate in refueling and can further improve the accuracy and robustness of the system by integrating the navigation information of radar, infrared, and other multi-sensor fusions.

**Author Contributions:** Conceptualization, K.G. and Y.X.; methodology, K.G.; software, X.X.; validation, K.G., X.X. and J.R.; formal analysis, K.G. and Y.H.; data curation, X.X.; writing—original draft preparation, K.G.; writing—review and editing, K.G., Z.Z. and Y.X.; visualization, X.X.; supervision, B.L. and Z.Z.; project administration, B.L. and Y.X.; funding acquisition, Y.X. All authors have read and agreed to the published version of the manuscript.

**Funding:** This research was funded by Natural Science Basic Research Program of Shaanxi (D5110220135).

**Data Availability Statement:** Data are available with the first author and can be shared with anyone upon reasonable request.

**Acknowledgments:** I would like to thank the China Academy of Aeronautics for providing me with a platform for conducting experiments.

**Conflicts of Interest:** The authors declare no conflict of interest.

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
