# Peer review of "Research of an Unmanned Aerial Vehicle Autonomous Aerial Refueling Docking Method Based on Binocular Vision"

_drones, doi:10.3390/drones7070433_

Round 1
Reviewer 1 Report
In this paper, the authors consider the navigation problem during autonomous aerial refueling docking of UAVs, using a lightweight detection model in order to improve the high accuracy and speed of the mission. The results contain some interesting elements, however, the following aspects should be paid more attention and revised accordingly:
1、 In the abstract, a lightweight drogue detection model specifically using image alignment and depth-separable convolution is proposed, but depth-separable convolution is not found in the text.
2、 In the second paragraph of the introduction, it states that there are hard-tube and hose-tube, but you only state the flow of the hose-tube, because the hard-tube are not applicable to the algorithm of this paper? Is it possible to add some explanations?
3、 Some of the diagrams in Figure 4 look very blurry, is it possible to redraw them?
4、 The paper mentions the marker points several times, but observe that the picture is some small green circles, so how are the positions of these small circles obtained, and when solving for the drogue pose, is only the center point of these marker points used, then should the radius of the small circles be added?
5、 The proposed pose estimation method in this paper, obtain the positions of marker points by binocular vision, then use these three marker points to obtain the center position of drogue, further obtain the transformation of the camera coordinate system and the drogue coordinate system, and final use the visual inverse projection to optimize the pose. But how is the visual inverse projection used to optimize the pose?
6、 The references in this paper are not adequate or up-to-date. More recent and highly relevant papers should be added and compared to show more clearly the new features of this paper.
N/A
Reviewer 2 Report
The paper presents a visual navigation method for autonomous aerial refueling docking using binocular vision and deep learning techniques. The proposed method includes a lightweight drogue detection model, a pose estimation algorithm, and simulation experiments. Overall, the paper addresses an important problem in unmanned aerial vehicle (UAV) navigation and provides promising results. However, there are a few areas that require further clarification and improvement before considering this paper for publication. Below are my specific comments:
* The spatial geometric modeling using optical markers for estimating the pose of the drone is an interesting approach. However, it would be helpful to elaborate on the optical markers' characteristics, placement, and their role in improving the accuracy and robustness of the algorithm. * The article claims an improvement of 1.18% in positioning accuracy compared to existing advanced methods. However, it would be helpful to know the baseline methods used for comparison and the statistical significance of the results. Please provide more details about the comparison experiments and statistical analysis conducted. Please provide these details to ensure the reproducibility and reliability of the results.
* The proposed drone detection model achieves a mean average precision (mAP) of 98.23% and a detection speed of 41.11 FPS in the embedded module. It would be beneficial to understand the trade-off between accuracy and speed in the context of other comparable methods. Please include a comparison of the proposed model's performance with existing state-of-the-art models.
* The paper lacks a discussion on the limitations of the proposed method. It is important to address the potential challenges and scenarios in which the method may not perform optimally. Furthermore, suggesting potential future directions for improvement and extension of the proposed work would strengthen the impact of the research. Please provide a dedicated section discussing the limitations and future work.
* The experimental setup section lacks some crucial details. It is essential to provide a clear description of the dataset used for training and evaluation, including the size, characteristics, and diversity of the data. Additionally, information about the hardware platform used, such as the UAV specifications and the embedded module details, would be helpful to understand the feasibility and scalability of the proposed method.
The only other comment for the authors it to please increase the quality of the images. Currently, many of the images in the article are very blurry.
Reviewer 3 Report
This manuscript presented a visual navigation method based on binocular vision and a deep learning approach to solve the navigation problem of the unmanned aerial vehicle autonomous aerial refueling docking process.
I have several comments as follows,
- figures should be used of higher quality (eg. fig 1)
- figure captions should be explained in detail their content
- Please read and fix typos and grammar problems (ex. line 106,...)
- Figure 3, which did not show the input and output of each block( what signal, data ...), the here relative pose was not explained or ref.
- mathematic equations were not referred
- 4.3. Visual navigation experiment, that did not show the trajectory( 2D and 3D)
- Please compare to other visual-based odometry methods ( ORB_SLAM, VINS-FUSION,...
- Please review more related works to VINS likes,
[1] Huang, Guoquan. "Visual-inertial navigation: A concise review." 2019 international conference on robotics and automation (ICRA). IEEE, 2019.
[2] Nam, D.V.; Gon-Woo, K. Robust Stereo Visual Inertial Navigation System Based on Multi-Stage Outlier Removal in Dynamic Environments. Sensors 2020, 20, 2922. https://doi.org/10.3390/s20102922
[3] Qin, T.; Li, P.; Shen, S. VINS-Mono: A robust and versatile monocular visual-inertial state estimator. arXiv 2017, arXiv:1708.03852.
N/A
Round 2
Reviewer 1 Report
The authors have revised the manuscript and I have no further comments.
The authors have revised the manuscript and I have no further comments.
Author Response
Thank you for your letter and for the Reviewers' comments concerning our manuscript entitled "Research of an unmanned aerial vehicle autonomous aerial re-fueling docking method based on binocular vision (Manuscript Number: drones-2433915)". Those comments are all valuable and very helpful for revising and improving our paper, as well as the essential guiding significance to our research.
Regards
Kun Gong
Reviewer 3 Report
Thank you for your efforts,
Still, I saw several problems that need to be addressed:
- Figures should be explained in detail in the caption
- Please fix typos errors and Grammarly in detail
- All mathematic equations (not yours) need to be cited
- In Fig 3, while the middle block shows a straightforward solution, the right block only shows in general.
- Subsection 3.4.3: The visual reprojection-based positional optimization method was not presented in detail (how to solve, tool, ...) and also not compared in the experimental.
- The distance to 3D markers was not indicated in the experiment or theory; I believe it has an essential effect on accuracy.
- The most contribution is using deep learning to detect the marker. I think training on the real-world dataset affects the results; please consider the real-world dataset.
please check in detail
Round 3
Reviewer 3 Report
The authors addressed my concerns,
Thank you for your efforts, and I have no further comments
N/A